# Ultrafast quantum control of ionization dynamics in krypton

Konrad Hütten[1,2], Michael Mittermair[1,2], Sebastian O. Stock[3,4], Randolf Beerwerth [3,4], Vahe Shirvanyan[1,2], Johann Riemensberger[1,2], Andreas Duensing[1], Rupert Heider[1], Martin S. Wagner[1], Alexander Guggenmos [2], Stephan Fritzsche[3,4,5], Nikolay M. Kabachnik[6,7,8], Reinhard Kienberger[1,2] & Birgitta Bernhardt [1,5,9]

Ultrafast spectroscopy with attosecond resolution has enabled the real time observation of ultrafast electron dynamics in atoms, molecules and solids. These experiments employ attosecond pulses or pulse trains and explore dynamical processes in a pump–probe scheme that is selectively sensitive to electronic state of matter via photoelectron or XUV absorption spectroscopy or that includes changes of the ionic state detected via photo-ion mass spectrometry. Here, we demonstrate how the implementation of combined photo-ion and absorption spectroscopy with attosecond resolution enables tracking the complex multi-dimensional excitation and decay cascade of an Auger auto-ionization process of a few femtoseconds in highly excited krypton. In tandem with theory, our study reveals the role of intermediate electronic states in the formation of multiply charged ions. Amplitude tuning of a dressing laser field addresses different groups of decay channels and allows exerting temporal and quantitative control over the ionization dynamics in rare gas atoms.

[1] Physics Department E11, Technical University of Munich, Garching 85748, Germany. [2] Max Planck Institute of Quantum Optics, Garching 85748, Germany. [3] Helmholtz-Institut Jena, Jena 07743, Germany. [4] Theoretisch-Physikalisches Institut, Friedrich Schiller University Jena, Jena 07745, Germany. [5] Abbe Center of Photonics, Friedrich Schiller University Jena, Jena 07745, Germany. [6] European XFEL GmbH, Hamburg, Schenefeld 22869, Germany. [7] Skobeltsyn Institute of Nuclear Physics, Lomonosov Moscow State University, Moscow 119991, Russia. [8] Donostia International Physics Center (DIPC), San Sebastian/Donostia E-20018, Spain. [9] Institute of Applied Physics, Friedrich Schiller University Jena, Jena 07745, Germany. Correspondence and requests for materials should be addressed to B.B. (email: Birgitta.Bernhardt@uni-jena.de)

In the last decade, time-resolved spectroscopy with attosecond resolution[1–13] has revolutionized our understanding of electron dynamics by capturing ultrafast processes in atoms, molecules and solids in real time. While the first photoelectron studies with attosecond resolution could, for example, track a few-femtosecond Auger process[14], meanwhile ultrafast metrology became so advanced that unexpected delays in photoemission from atoms were discovered[4,15], most recently down to sub-attosecond precision[5]. Attosecond time-resolved mass spectroscopy[16,17] provided the first time-domain observation of field-induced tunnel ionization manifesting itself in a step-like rise in the ion yield. More recently, attosecond transient absorption detection could launch and detect valence electron wavepackets in atoms[11], examine their interference with continuum states[13] and enabled studying tunnel ionization in solids[18].

Photoelectron detection techniques generally suffer from electron backgrounds produced by strong laser fields or secondary electrons that do not carry any information about the dynamics under inspection. The detection of the correspondingly produced ions, however, is background-free. Nevertheless, both methods require the release of photoelectrons and thus are suitable only for the study of ionizing events. Transient absorption spectroscopy (TAS) in contrast is not limited to phenomena liberating electrons and is the method of choice for investigating bound–bound transitions[19,20]. Instead of measuring the yield of generated charge carriers, TAS measures the spectrally resolved absorption of an attosecond extreme ultraviolet (XUV) pulse in a medium that has been coherently excited by an XUV pulse or dressed by an intense, time-delayed near-infrared (NIR) few-cycle laser pulse. This method typically provides a higher resolution in energy ($\Delta E/E \sim 10^{-3}$) than ion or electron detection ($\Delta E/E \sim 10^{-2}$ at best). However, it generally lacks the dynamic range to simultaneously detect the characteristic absorption signals of different co-existing ionic charge states due to insufficient spectral bandwidth or due to the absorption cross-sections and abundances of the different ionization states that typically vary by several orders of magnitude. TAS experiments have been successful in recording auto-ionizing state lifetimes in xenon[12] and other species[21]. However, even a detailed theoretical investigation could not explain whether the interrogating ultrashort and intense NIR laser pulse couples the auto-ionizing states to neighboring resonances or to which extent excited electrons are promoted into the ionization continuum by the laser field[22]. In contrast, the ion detection that enabled the first observation of tunneling electrons in neon and xenon could track the change in ion yield of different charge states. However, mass spectroscopic studies to this date cannot resolve individual shake up satellites[16].

To overcome the limitations of single observable experiments and in order to draw a complete picture of the co-evolving excitation/ionization dynamics, in this article we demonstrate the benefit of combining ion spectroscopy and transient absorption with attosecond temporal resolution. Merging these detection methods provides complementary insight into the excitation/decay mechanism as the transient absorption maps the initiating resonant excitation and the ion detection sensitively records the subsequent branching into intermediate and final states. The combination of transient absorption with ion spectroscopy compensates the limits in the simultaneous detection of several ionic states that absorption spectroscopy typically brings along. The prerequisite is that the pump–probe experiment can be performed twice under identical experimental conditions except for differing target densities optimized for the two detection methods. Adjusting the laser electric field amplitude allows selectively addressing different intermediate states of the auto-ionization cascades that follow the XUV excitation resulting in different apparent lifetimes. With that, the experimentalist

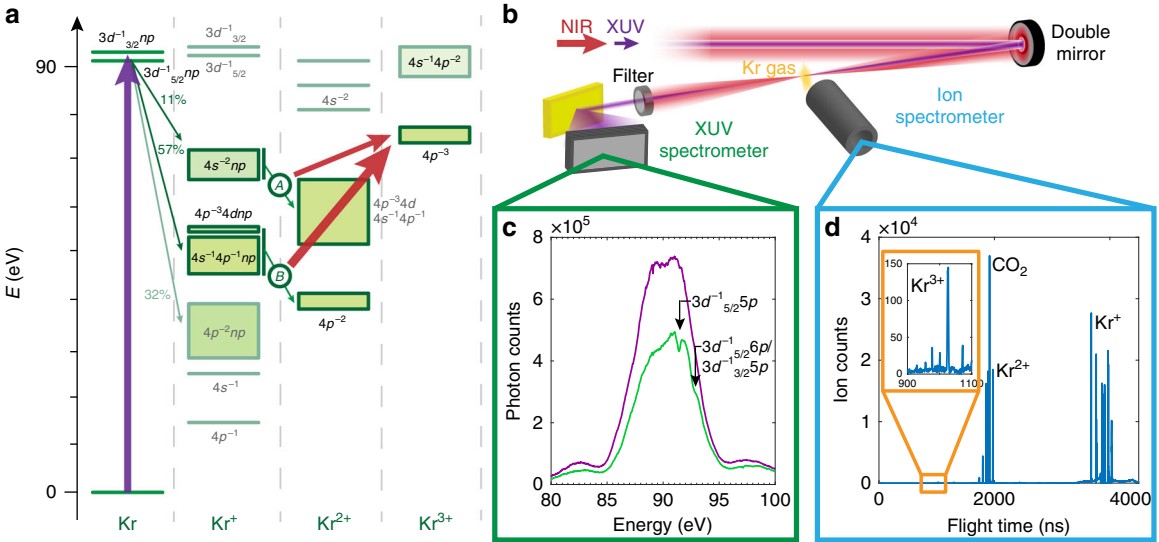

**Fig. 1** Overview of the experiment. **a** Krypton excitation and decay scheme. The XUV pulse excites the $3d^{-1}_{5/2}\,np/3d^{-1}_{3/2}\,np$ states (violet arrow) that can decay via different Auger cascades (green arrows). Depending on the NIR laser intensity, the different intermediate $Kr^+$ states can be further ionized to $Kr^{3+}$: at low NIR intensities, the intermediate $4s^{-2}np$ states can be ionized to $Kr^{3+}$ (thin red arrow, $I_{NIR} = (8.6 \pm 1) \times 10^{13}$ W cm$^{-2}$), at higher NIR intensities, the $4p^{-3}4dnp/4s^{-1}4p^{-1}np$ states can be ionized to $Kr^{3+}$ (thick red arrow, $(2.9 \pm 0.5) \times 10^{14}$ W cm$^{-2}$). The direct XUV ionization and the corresponding decay channels are not shown for the sake of clarity, but are considered in the data analysis. For a more detailed figure see refs.[17,31] for example. **b** Experimental setup with a double mirror configuration introducing a time delay between the XUV pulse and the NIR few-cycle pulse (the XUV pulse arriving first for positive time delays). XUV and NIR beams are focused into a krypton gas cloud. The remaining XUV radiation transmitted by the Kr gas is measured by an XUV spectrometer, while the Kr ions are detected by a reflectron-type ion spectrometer. **c** Incident XUV spectrum centered at 90 eV (violet) and a typical krypton transmission spectrum (green) showing the $3d^{-1}_{5/2}\,5p$ transition at 91.23 eV and the $3d^{-1}_{5/2}\,6p/3d^{-1}_{3/2}\,5p$ transitions at 92.45 eV, respectively. **d** Measured ion spectrum yielding singly, doubly and triply charged krypton ions. For the absorption and ion spectra of **c** and **d**, the NIR pulse was set to advance the XUV pulse by 150 fs ($\Delta t = -150$ fs)

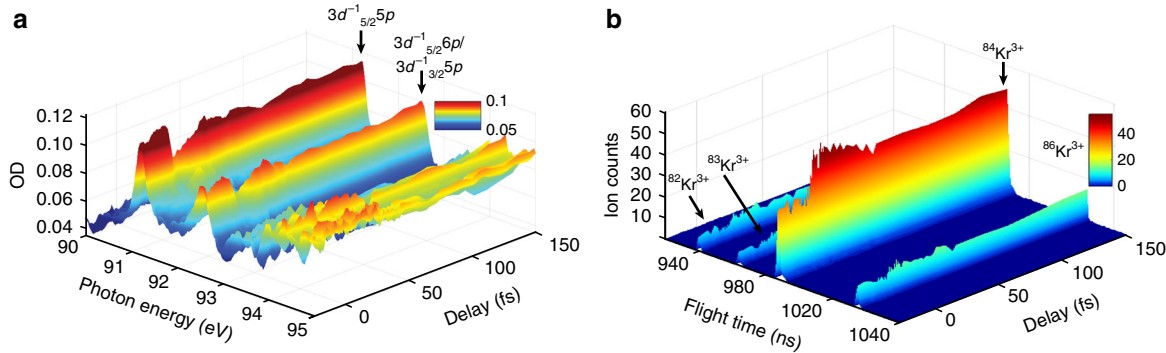

**Fig. 2** Absorption and ionization spectroscopy measurements. **a** Spectrally resolved optical density OD with respect to the time delay between the XUV and the NIR pulse, revealing the $3d^{-1}_{5/2}5p$ and $3d^{-1}_{5/2}6p/3d^{-1}_{3/2}5p$ resonances at 91.23 eV and 92.45 eV, respectively. The XUV pulse is preceding the NIR pulse for positive times on the delay axis. The absorbance at the resonances transiently decreases at XUV/NIR pulse overlap and subsequently recovers almost to its original value, with an exponential time constant corresponding to the state lifetimes. **b** Time-dependent and isotope-resolved $Kr^{3+}$ ion yield. For all isotopes similarly, the $Kr^{3+}$ ion yield rises shortly before XUV/NIR pulse overlap and decays with a slower time constant when compared to the transient absorption in the left panel to a persistent elevated count rate. In both cases, the NIR intensity was $(8.6 \pm 1) \times 10^{13}$ W cm$^{-2}$

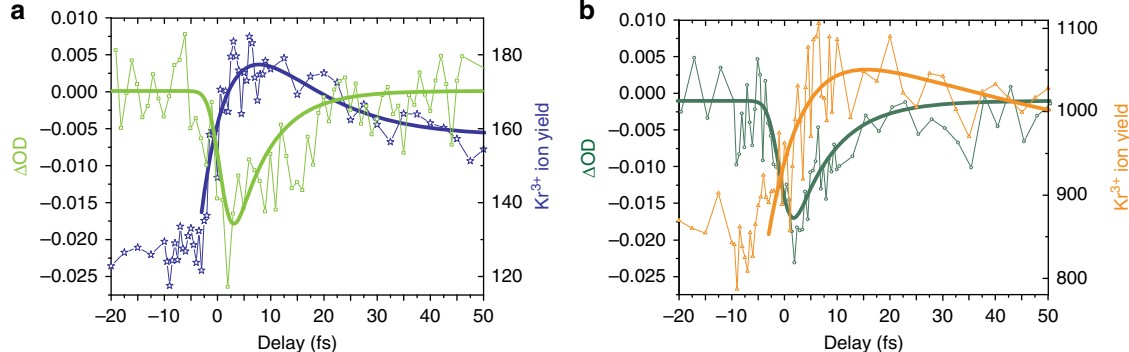

**Fig. 3** Absorbance change ΔOD and $Kr^{3+}$ ion yield vs. time delay. The change in absorbance or optical density ΔOD states the difference of the optical density OD $= -\log_{10}(I_t(\Delta t)/I_0)$ at a given time delay $\Delta t$ ($I_t(\Delta t)$ is the transmitted signal at time delay $\Delta t$, $I_0$ is the reference intensity measured at a time delay $t_R = -30$ fs). **a** At lower NIR intensities of $(8.6 \pm 1) \times 10^{13}$ W cm$^{-2}$, the absorbance (light green, left scale) transiently drops around pulse overlap, while the $Kr^{3+}$ ion yield (blue, right scale) shortly rises before it settles to an elevated ion yield of almost 160 counts in 20 s integration time. **b** At a NIR intensity three times as high, $(2.9 \pm 0.5) \times 10^{14}$ W cm$^{-2}$, the absorbance (dark green, left scale) transiently drops around pulse overlap similar to **a**, while the $Kr^{3+}$ ion yield (orange, right scale) shortly rises before it settles to an elevated ion yield of about 860 counts in 20 s integration time with a slower decay constant when compared to **a**. Please see Supplementary Note 2 for longer scans

obtains control of the temporal evolution and the absolute yield of the ionization dynamics by accessing different level groups in the cascade with different NIR intensities. The accompanying absorption measurement enables the determination of the instrument response function, the lifetimes of resonantly excited states involved in the process and proves that the overall conditions of the experiment (resonant excitation and ionization) remain unchanged for a large range of NIR intensities.

## Results

**Experiment**. To explore the decay dynamics of highly excited krypton (see Fig. 1a), a phase-stabilized Ti:Sapphire few-cycle laser[14,16,17] is used to produce isolated attosecond pulses (cp. Supplementary Fig. 1) via high harmonic generation in a pump–probe scheme united with a reflectron-type ion spectrometer and an XUV grating spectrometer (see Fig. 1b and Supplementary Note 1 for details). Figure 1c shows the transmission spectrum without (violet) and with (green) krypton gas sample with the NIR pulse preceding the XUV pulse by 150 fs. For the same time delay, Fig. 1d depicts the corresponding ion spectrum with singly, doubly and triply charged krypton ions. For the time-resolved studies, we measured the krypton transmission and ion

spectrum scanning the arrival time difference between XUV and laser pulse.

**Merging ultrafast absorption and ion mass spectroscopy**. Figure 2a shows the spectrally resolved change in absorbance (optical density OD $= -\log_{10}(I_t(\Delta t)/I_0)$), with $I_0$ as incident XUV spectrum) as function of the XUV/NIR time delay, while Fig. 2b depicts the isotope-resolved $Kr^{3+}$ ion yield change. The absorbance starts to transiently decrease at XUV/NIR pulse synchrony while the $Kr^{3+}$ ion yield shortly rises, however, with a slightly retarded response with respect to the absorption change.

**Ionization dynamics in krypton**. To reveal further details of the ionization dynamics, Fig. 3 presents the absorbance ΔOD at 91.23 eV (corresponding to the $3d_{5/2}5p$ resonance[23]) relative to the OD measured at $\Delta t = -30$ fs and the $Kr^{3+}$ ion yield added up over the four most abundant krypton isotopes for two different NIR intensities: Fig. 3a for low NIR intensity of $(8.6 \pm 1) \times 10^{13}$ W cm$^{-2}$ and Fig. 3b for high NIR intensity of $(2.9 \pm 0.5) \times 10^{14}$ W cm$^{-2}$. For both cases, the absorbance decreases on a quick (<10 fs) time scale before it recovers almost to its original value. The time constant of the fast ΔOD decrease reflects the

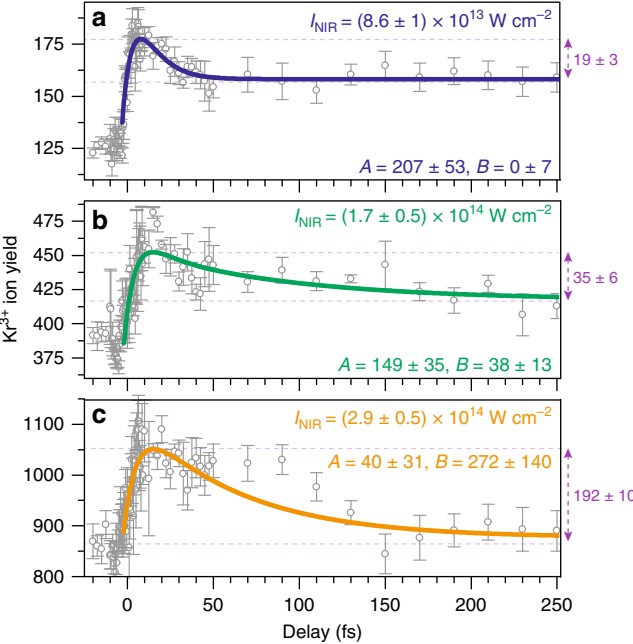

**Fig. 4** Control of Kr ionization dynamics. Time-dependent $Kr^{3+}$ ion yield for three different NIR intensities: **a** $I_{NIR} = (8.6 \pm 1) \times 10^{13}$ W cm$^{-2}$, **b** $I_{NIR} = (1.7 \pm 0.5) \times 10^{14}$ W cm$^{-2}$ and **c** $I_{NIR} = (2.9 \pm 0.5) \times 10^{14}$ W cm$^{-2}$. While for all three intensities, the rise in the $Kr^{3+}$ ion yield obeys the 7.9 fs $3d^{-1}np$ lifetime, the subsequent decrease of the ion counts is strongly affected by the NIR intensity: a double exponential least squares fit yields the decay constants of $\tau_A = 9.3 \pm 3.7$ fs and $\tau_B = 60 \pm 26$ fs with varying contributions in their amplitudes for the three intensities: **a** $A = 207 \pm 53$, $B = 0 \pm 7$, **b** $A = 149 \pm 35$, $B = 38 \pm 13$ and **c** $A = 40 \pm 31$, $B = 272 \pm 140$. This change of the amplitude ratio $B/A$ describes well the expected case that a second channel is starting to be addressed for elevated NIR intensities (see the levels marked with the green capital letters $A$ and $B$ in Fig. 1a and the text for details). For the highest intensity, an accompanying pulse of the ultrashort NIR laser pulse increases the ion yield at around 70–120 fs time delay (see Supplementary Note 3 for more details). Please note that the scaling, the initial and final values of the ion yields differ significantly for the three NIR intensities. The violet numbers on the right side of each panel indicate the difference between the maximum of the fitting function and the value the fitting function relaxes to for large time delays (see Supplementary Note 4 for a detailed explanation for the different ion yield values). The error bars show the standard error of the average of six measurements

instrument response and the NIR pulse duration. A least squares fit gives an instrument response function[24] of $1.8 \pm 0.4$ fs and hence a duration of $4.5 \pm 1$ fs for the Gaussian-shaped NIR pulse while the exponential recovery of the absorbance after pulse overlap yields a $6.8 \pm 1.5$ fs and a $7.7 \pm 2.2$ fs auto-ionization state lifetime of the $3d^{-1}np$ levels for low and high NIR intensity, respectively. These lifetime results agree well with the previously reported value of $7.9 \pm 0.2$ fs[25]. Potential NIR intensity-dependent effects like the ac Stark shift or resonant coupling to neighboring states resulting in a broadening or even splitting of the resonances could be excluded by careful investigation of the resonance center positions and line shapes for all recorded time delays. This and the comparison to the literature values confirm that the initial Auger decay is virtually unaffected by different NIR laser field intensities. Recovery of the $Kr^{3+}$ yield shows a strong dependence on the intensity of the applied NIR laser field: the number of $Kr^{3+}$ ions increases when the XUV and the NIR pulses start to overlap, but not as quickly as the absorption changes at the $3d^{-1}np$ resonances. This is because at pulse overlap, a significant part of

the $Kr^{3+}$ ions result from a cascaded Auger decay following the resonant $3d^{-1}np$ excitation. Hence, the $Kr^{3+}$ ion yield rise time does not only depend on the NIR pulse duration but also on the lifetimes of the resonantly excited $3d^{-1}np$ states. Taking into account the instrument response function and the $3d^{-1}np$ lifetime for the increase of the $Kr^{3+}$ ion yield curve, a least squares fit yields a time constant of $9.3 \pm 3.7$ fs for the tail of the $Kr^{3+}$ ion count rate that corresponds to the effective lifetime of the intermediate state of the Auger cascade (here: $I_{NIR} = (8.6 \pm 1) \times 10^{13}$ W cm$^{-2}$). This value differs from the previously reported value of $20 \pm 4$ fs[17]: our presented study is to our knowledge the first one that systematically investigates the ionization dynamics in krypton for different NIR intensities. As the lowest NIR intensity is twice as high as previously[17], different auto-ionization paths in the de-excitation of the $Kr^{+}$ ions become visible and may lead to a different apparent/effective decay time. In practice, we here observe an effective lifetime for the $3d^{-1}np \rightarrow 4s^{-2}np \rightarrow 4s^{-1}4p^{-1}/4p^{-3}4d$ cascades that arise from a large number of individual but not resolved fine-structure transitions. This is confirmed by our multiconfiguration Dirac–Fock calculations using the GRASP[26] and RATIP[27] codes and by taking the average of all the individual lifetimes, weighted by their relative decay probabilities. This theoretical approach is for the first time applied to reveal effective lifetimes of different potential group cascades (see Supplementary Note 5 for details).

**Control of krypton ionization dynamics.** Figure 4 shows for three different NIR intensities that the observed decay time increases for higher NIR intensities due to a second set of decay channels ($3d^{-1}np \rightarrow 4s^{-1}4p^{-1}np/4p^{-3}4dnp \rightarrow 4p^{-2}$). This second set (indicated by $B$ in Fig. 1a) has a much longer effective lifetime as it was already speculated[17]: A double exponential decay fit gives rise to a second decay time of $60 \pm 28$ fs, if we apply 9.3 fs as the first decay time that has been found for the lowest intensity. At higher NIR intensity, channel $B$ results in an increase of the amplitude ratio $B/A$. Amplitude $A$ corresponds to the Auger cascade emerging at low NIR intensities. The results of our theoretical calculations of the effective lifetimes with 6 fs for the $4s^{-2}np$ levels and 49 fs for the $4s^{-1}4p^{-1}np/4p^{-3}4dnp$ levels agree very well with the measurements.

There occurs a transient drop in the $Kr^{3+}$ ion production shortly before pulse overlap (approximately at −5 fs) that depends on the NIR intensity but appears independent of the carrier envelope phase (strongest dip at high NIR intensities, best visible in Fig. 3b). This dip has been observed previously[17], and its origin is not yet understood. It could result from a transient population transfer[28] but may arise also as a Fano-type resonance[29] that is embedded into the $Kr^{2+}$ continuum.

## Discussion

While a control of the end configuration in the dissociative ionization process of deuterium by tuning the carrier envelope phase has been already demonstrated[30], we have shown here that combined attosecond transient absorption and mass spectrometry allows the observation and control of the XUV-induced ionization dynamics in rare gas atoms. The study reveals the role of intermediate electronic and ionic states and highlights how laser-induced state coupling can be used to control the post-excitation decay dynamics. The experiment simultaneously determines the instrument response function, the effective lifetimes of the resonantly excited states and confirms that the initiating resonant excitation is not affected by the dressing laser field amplitude while the evolution of the post-excitation decay can be dynamically and quantitatively manipulated.

**Data availability**. The data that support the findings of this study are available from the corresponding author upon request.

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

## Acknowledgements

The authors warmly acknowledge Hartmut Schröder and Martin Schultze for instrumental support and discussions, as well as Marinus Mayerbacher's and Alexander Späh's contributions to the vacuum setup. B.B. wants to thank the Alexander von Humboldt foundation and the Carl Zeiss foundation for financial support. N.M.K. acknowledges hospitality and financial support from DIPC (San Sebastian/Donostia) and from the theory group in cooperation with the SQS work package of European XFEL (Hamburg). R.K. acknowledges a Consolidator Grant from the European Research Council (ERC-2014-CoG AEDMOS). S.O.S. gratefully acknowledges funding by the German Federal Ministry of Education and Research (BMBF) under Contract No. 05K16SJA. This work was supported by the Max Planck Society, the Deutsche Forschungsgemeinschaft Cluster of Excellence: Munich Centre for Advanced Photonics (http://www.munich-photonics.de).

## Author contributions

Experimental studies and analysis of experimental and theoretical signatures were carried out by K.H., M.M., V.S. and B.B. Theory and modeling were performed by S.O.S. and R.B., supervised by S.F. Customized XUV optics were provided by A.G. J.R., A.D., R.H., M.S.W. and R.K. contributed to the experimental setup and measurement. The manuscript was written by N.M.K. and B.B. All authors discussed the results and commented on the paper.

## Additional information

**Competing interests:** The authors declare no competing financial interests.

