## [Peer Review File · Nature Communications]

Reviewers' comments:

Reviewer #1 (Remarks to the Author):

Title:

"Ultrafast Quantum Control of Ionization Dynamics"

Nature Communications manuscript NCOMMS-17-23505-T

Hütten and co-workers present the first measurements of complex excitation and cascaded Auger decay by combined transient absorption spectroscopy (TAS) and photo-ion charge/mass detection. The ionization processes in Kr are initiated by an isolated attosecond pulse and subsequently probed by an ultra-short laser pulse. Control of the complex ionization processes by laser amplitude tuning is shown. The question arises as to how this combined technique brings new insight to the complex ionization dynamics.

Major points:

(1) In the introduction the authors write that this is the "first implementation of combined photo-ion and absorption spectroscopy with attosecond resolution", but no results on the attosecond time scale are presented in the article. The shortest duration reported is the experimental instrument response time of 1.8 fs. For this reason, I think that the word "ultrafast", which implies time-scales down to few femtoseconds, should be used in the introduction and throughout the manuscript instead of "attosecond". I note that the correct terminology is already used in the title "Ultrafast Quantum Control of Ionization Dynamics".

(2) The authors write that TAS lacks the "dynamic range" to resolve the characteristic absorption signals for different ions. Could the authors elaborate on what they mean with "dynamic range" of TAS: is it a fundamental problem of TAS or a practical problem due to insufficient bandwidth or intensity of the attosecond pulse? This is an important point for the major claim of the manuscript and it should be made more clear also in the conclusions. As an outlook: Would it be possible in future experiments to measure resonances of generated ions by TAS or will it always be important to rely on charge/mass detection?

(3) In Fig.3 the optical density is shown for TAS measurements with two different laser probe field intensities. While, I agree that the fit with the 7.9 fs looks "OK" for the low intensity case in panel a, I do not agree that the fit is as good for the higher intensity case in panel b. This is an important point that is stressed in the conclusions: "the initiating resonant excitation is not affected by the dressing laser field amplitude". Since this is a major point, I ask: Why are only the curves corresponding to 7.9 fs shown and not curves fitted to the TAS experimental data to extract the actual life times of the two different intensities? The extracted values for the two intensities should be reported (with error bars) so that they can be compared with the "expected" value 7.9 fs. Could the authors also comment on possible reasons for why the lifetime could depend on the laser probe intensity? Did the authors see any change in the position of the $3d_{5/2}$ $5p$ resonance with laser intensity?

A few minor points are listed below.

Minor points:

(i) If the authors wish to cite a theoretical paper on laser-assisted attosecond photoemission, Reference 2 is not the most up to date article to cite since it does not include the important laser-Coulomb coupling term of the streak camera technique. A better choice would be:

Attosecond streaking:

Faraday Discuss., 2013, 163, 353

(ii) Transient absorption is not "independent of free electrons" as is claimed in the manuscript. The polarization of bound electron wave packets to the free continuum will lead to control of absorption but this effect has so far only been studied theoretically:

<http://iopscience.iop.org/article/10.1088/2040-8986/aa8a93>

(iii) Line 110 the reaction may end up at " $4s^{-1} 4p^{-1}$ " not " $4p^{-1} 4p^{-1}$ ".

(iv) Line 116 "->" should be replaced with right arrows.

(v) In Fig.1 (c) the text "XUV,IR" with arrows below implies to me that the NIR arrives before the XUV because (because the pulses move to the right in the figure). Also in the text "delay between XUV and NIR" implies that $\text{delay} = \tau_{\text{XUV}} - \tau_{\text{NIR}}$, which is not consistent with the sign convention of the delay used: "The XUV pulse is preceding the NIR pulse for positive times on the delay axis". These two points made me feel unsure of the order of the pulses when reading the manuscript for the first time. While this may be obvious to experts in the field that the NIR should come after the XUV, such small inconsistencies may cause trouble for general readers.

Overall, I think that the work is impressive and that manuscript is well-written. Given that my three major points are cleared up I support that the manuscript can be published as an article in Nature Communications.

/ Jan Marcus Dahlström

Reviewer #2 (Remarks to the Author):

Konrad Hütten et al performed a XUV pump NIR probe study to investigate the electronic dynamics in krypton atoms following XUV excitation. Photo-ion and attosecond transient absorption spectroscopies are combined in a single experiment aiming to explore complex excitation and ionization processes simultaneously. Such combined technique has demonstrated its advantage of extracting multitude of information including the instrument response function, lifetime of autoionizing states

as well as the intensity dependence of the multi-step Auger decay processes. After the XUV excitation, the highly excited states $3d-1np$ experiences a two step Auger decay process until the NIR pulse arrives to terminate the relaxation steps. The attosecond transient absorption measurement reveals the lifetime of the first Auger decay process, and the second step Auger decay dynamics is tracked by the photoion measurement. This is an appealing work that holds the promise to access complete dynamic information during light-matter interaction, which is normally inaccessible using single spectroscopy. I recommend its publication in Nature Communication with the following comments addressed.

- 1) Although combining different spectroscopies can potentially lead to richer information, compatibility between different spectroscopies is a prerequisite. For example, the photo-ion/photoelectron spectroscopy prefers dilute target to avoid the space charge effect, a successful combined technique is the Reaction Microscope, which combines the photoelectron and photoion spectroscopy in coincidence. The attosecond transient absorption spectroscopy typically requires a target density that is several orders of magnitude higher than that for the photoelectron/photoion spectroscopy in order to create observable signatures in the transmitted spectrum. This incompatibility indicates that the reported technique is limited to processes that do not have severe ionization, therefore extending the technique to more complex molecular systems with lower ionization threshold, is probably challenging. It will be valuable to include some discussions on this issue.
- 2) The time constants and the retrieved branching ratio of the Auger decay channels in Fig4b and Fig4c may not be reliable. The humps appearing around 50fs are detrimental for accurate fittings. When humps are getting amplified at higher NIR intensity, it is rather difficult to retrieve a meaningful ratio of A/B. The authors should clarify this since this is the major discovery of the current work.
- 3) The XUV spectrometer energy resolution is energy dependent; it should be specified at which energy the resolution is 100 meV.
- 4) In line 110 of page 7: the final state for the cascade should be " $4s^{-1} 4p^{-1}/4p^{-3}4d$ "

Reviewer #3 (Remarks to the Author):

In this manuscript the authors study Auger cascade in Krypton by means of time-resolved techniques, i.e. the Transient Absorption Spectroscopy (TAS) and the photo-ion mass spectroscopy. They found that the TAS gives results in reasonable accordance with the autoionizing process envisioned, however the ion signals indicate the presence of an Auger cascade which lifetime strongly depends on the NIR laser intensity. The combination of these two techniques is rather a "tour de force" and the results are interesting, but I do not think this manuscript deserves publication in Nature Communications.

1/ I consider that this manuscript is a follow-up of a paper [13] already published by the same authors. The only differences concern the use of the combined techniques but it is not clear at all, what are the new physical insights gained here.

2/ There is no real answer proposed on the influence of the NIR intensity on the lifetime in the Auger cascade. Different options are presented but I believe the authors should give more detailed reasons.

3/ Most figures are very difficult to read and should be redesigned to give useful informations to the reader. For example, the choice of yellow in Fig. 1a is « awful » and Figs. 2 cannot be fully exploited (the traces of 82 and 83 Kr³⁺ are invisible). I suggest to remove Figs. 2 from the manuscript and put them in the SI. This would free more space for the authors to give more details.

4/ I am very puzzled by the results shown in Fig. 4. It might be an optical illusion but the decrease of Fig.4b seems slower or with the same slope than in Fig.4c, while the ratio A/B are totally different. Could the authors comment on that? Furthermore, I do not understand the notion of post pulse put forward to explain the increase around 80fs. More informations to the reader would be welcome on its origin and why it is not present at lower intensities and what would be the options to remove it.

5/ The authors claim they have a coherent control on the decay channels but I would not call it « coherent » as it does not depend on the delay between the XUV and the NIR nor on the NIR phase. It is only a « intensity » effect, from my point of view.

Reviewer #1 (Remarks to the Author):

Title:

"Ultrafast Quantum Control of Ionization Dynamics" Nature Communications manuscript NCOMMS-17-23505-T

Hütten and co-workers present the first measurements of complex excitation and cascaded Auger decay by combined transient absorption spectroscopy (TAS) and photo-ion charge/mass detection. The ionization processes in Kr are initiated by an isolated attosecond pulse and subsequently probed by an ultra-short laser pulse. Control of the complex ionization processes by laser amplitude tuning is shown. The question arises as to how this combined technique brings new insight to the complex ionization dynamics.

Major points:

- (1) In the introduction the authors write that this is the "first implementation of combined photo-ion and absorption spectroscopy with attosecond resolution", but no results on the attosecond time scale are presented in the article. The shortest duration reported is the experimental instrument response time of 1.8 fs. For this reason, I think that the word "ultrafast", which implies time-scales down to few femtoseconds, should be used in the introduction and throughout the manuscript instead of "attosecond". I note that the correct terminology is already used in the title "Ultrafast Quantum Control of Ionization Dynamics".

The reviewer states correctly that in the presented experiment, time-resolved spectroscopy with attosecond resolution is used to observe ultrafast processes that occur on the few-femtosecond time scale. To make this more transparent in the manuscript, we changed the following sections accordingly:

In the introductory paragraph:

Ultrafast spectroscopy with attosecond resolution has enabled the real time observation of ultrafast electron dynamics... We demonstrate how the first implementation of combined photo-ion and absorption spectroscopy with attosecond resolution enables tracking the complex multidimensional excitation and decay cascade of an ultrafast Auger autoionization process of a few femtoseconds in highly excited rare gas atoms.

Throughout the manuscript (marked in red), we now specify the two techniques as ion spectroscopy and transient absorption with attosecond temporal resolution instead of "attosecond spectroscopy" in order to be accurate.

- (2) The authors write that TAS lacks the "dynamic range" to resolve the characteristic absorption signals for different ions. Could the authors elaborate on what they mean with "dynamic range" of TAS: is it a fundamental problem of TAS or a practical problem due to insufficient bandwidth or intensity of the attosecond pulse? This is an important point for the major claim of the manuscript and it should be made more clear also in the conclusions. As an outlook: Would it be possible in future experiments to measure resonances of generated ions by TAS or will it always be important to rely on charge/mass detection?

What we referred to as a lack of “dynamic range” of TAS is indeed a fundamental consideration in such measurements: Experimental parameters can be optimized to record the characteristic absorption features of a specific ionization state by tuning the central photon energy, bandwidth and flux of the XUV pulses used for detection. In a sample where several ionization states co-exist, their relative abundance typically varies by orders of magnitude due to the strong differences in fractional cross sections for single-photon (X)UV ionization or the highly nonlinear intensity dependence of (multiple-) ionization probabilities in multi-photon or tunneling ionization. Thus, for two different ionization states of the same atomic species to be detectable in the same TAS experiment, they both need to have characteristic absorption features within the bandwidth of the XUV pulses and have a similar number density – at least within a factor of 10^2 .

In contrast, the quantum efficiency of the ion spectrometer is almost 100 % regardless of the ionization state. This results in a high detection rate for all charge states generated directly by e.g. single or double ionization but also indirectly by decay mechanisms. Due to the flight time separation of different ionic states and the high quantum efficiency of the detection, there is virtually no upper limit to the ratio between two ionic states that can still be quantitatively distinguished. Experimentally a dynamic range of at least 10^9 is routinely achieved.

Combining transient absorption with ion spectroscopy compensates the limits that absorption spectroscopy typically brings along. Because this aspect has not been discussed comprehensively in our manuscript, we added to lines 53 and following of the originally submitted manuscript version:

However, it generally lacks the dynamic range to simultaneously detect the characteristic absorption signals of different co-existing ionic charge states due to insufficient spectral bandwidth or due to the absorption cross sections and abundances of the different ionization states that typically vary by several orders of magnitude. TAS experiments...

And to lines 64 and following:

Merging these detection methods provides complementary insight into the excitation/decay mechanism as the transient absorption maps the initiating resonant excitation and the ion detection sensitively records the subsequent branching into intermediate and final states. The combination of transient absorption with ion spectroscopy compensates the limits in the simultaneous detection of several ionic states that absorption spectroscopy typically brings along. The prerequisite is that the pump-probe experiment can be performed twice under identical experimental conditions except for differing target densities optimized for the two detection methods. Adjusting the laser electric field amplitude allows selectively addressing different intermediate states of the autoionization cascades that follow the XUV excitation resulting in different apparent lifetimes. With that, the experimentalist obtains control of the temporal evolution and the absolute yield of the ionization dynamics by accessing different level groups in the cascade with different NIR intensities.

- (3) In Fig.3 the optical density is shown for TAS measurements with two different laser probe field intensities. While, I agree that the fit with the 7.9 fs looks "OK" for the low intensity case in panel a, I do not agree that the fit is as good for the higher intensity case in panel b. This is an important point that is stressed in the conclusions: "the initiating resonant excitation is not affected by the dressing laser field amplitude". Since this is a major point, I ask: Why are only the curves corresponding to 7.9 fs shown and not curves fitted to the TAS experimental data to extract the actual life times of the two different intensities? The extracted values for the two intensities

should be reported (with error bars) so that they can be compared with the "expected" value 7.9 fs. Could the authors also comment on possible reasons for why the lifetime could depend on the laser probe intensity? Did the authors see any change in the position of the 3d_{5/2} 5p resonance with laser intensity?

Thank you for pointing out that our findings will gain in significance if independently extracted lifetime values are compared to the previously reported.

We performed least squares fits to our measurements, updated figure 3 accordingly, and added our conclusions concerning potential resonance shifting and broadening effects with the following paragraph, starting at line 95 of the original manuscript:

A least squares fit gives an instrument response function²⁵ of (1.8±0.4) fs and hence a duration of (4.5±1) fs for the Gaussian-shaped NIR pulse while the exponential recovery of the absorbance after pulse overlap yields a (6.8±1.5) fs and a (7.7±2.2) fs autoionization state lifetime of the 3d¹np levels for low and high NIR intensity, respectively. These lifetime results agree well with the previously reported value of (7.9±0.2) fs²⁶. Potential NIR intensity dependent effects like the ac Stark shift or resonant coupling to neighboring states resulting in a broadening or even splitting of the resonances could be excluded by careful investigation of the resonance center positions and line shapes for all recorded time delays. This and the comparison to the literature values confirm that the initial Auger decay is virtually unaffected by different NIR laser field intensities.

A few minor points are listed below.

Minor points:

- (i) If the authors wish to cite a theoretical paper on laser-assisted attosecond photoemission, Reference 2 is not the most up to date article to cite since it does not include the important laser-Coulomb coupling term of the streak camera technique. A better choice would be: Attosecond streaking: Faraday Discuss., 2013, 163, 353

We followed the advice of the reviewer and added the reference as [9] to the manuscript.

During our literature study, we identified an even more recent publication on that matter that we added as reference [10]: R. Pazourek et al. Review of Modern Physics Vol. 87, 765 (2015).

Due to its topicality, we added a recent work (published in September 2017) on the delay in photoemission that we mention on page 2 and that we cite as reference [17].

- (ii) Transient absorption is not "independent of free electrons" as is claimed in the manuscript. The polarization of bound electron wave packets to the free continuum will lead to control of absorption but this effect has so far only been studied theoretically: <http://iopscience.iop.org/article/10.1088/2040-8986/aa8a93>

Because our initial phrasing of "Transient absorption spectroscopy (TAS) in contrast does not depend on free electrons" is misleading (lines 48 and 49 in the original manuscript version), we modified it to "Transient absorption spectroscopy (TAS) in contrast is not limited to phenomena liberating electrons".

- (iii) Line 110 the reaction may end up at "4s⁻¹ 4p⁻¹" not "4p⁻¹ 4p⁻¹".

Thank you for finding this mistake that we eliminated.

(iv) Line 116 "->" should be replaced with right arrows.

We added "real" arrows accordingly.

(v) In Fig.1 (c) the text "XUV,IR" with arrows below implies to me that the NIR arrives before the XUV because (because the pulses move to the right in the figure). Also in the text "delay between XUV and NIR" implies that $\text{delay} = \tau_{\text{XUV}} - \tau_{\text{NIR}}$, which is not consistent with the sign convention of the delay used: "The XUV pulse is preceding the NIR pulse for positive times on the delay axis". These two points made me feel unsure of the order of the pulses when reading the manuscript for the first time. While this may be obvious to experts in the field that the NIR should come after the XUV, such small inconsistencies may cause trouble for general readers.

Thank you very much for pointing out this potential unclarity. In order to be more consistent, we changed the figure panel (after our review panel 1b) in accordance with your comment. We kept the expression "delay between XUV und NIR" in the text because in our opinion mentioning the XUV pulse first indicates that the XUV pulse arrives also first at the sample. In order to be more explicit on this definition, we added short explanations to the corresponding sections:

In the caption of figure 1: ... a time delay between the XUV pulse and the NIR few-cycle pulse (the XUV pulse arriving first for positive time delays).

In the caption of figure 2:

Spectrally resolved optical density OD with respect to the time delay between the XUV and the NIR pulse, revealing the $3d^{-1}_{5/2}5p$ and $3d^{-1}_{5/2}6p/3d^{-1}_{3/2}5p$ resonances at 91.23 eV and 92.45 eV, respectively. The XUV pulse is preceding the NIR pulse for positive times on the delay axis.

Overall, I think that the work is impressive and that manuscript is well-written. Given that my three major points are cleared up I support that the manuscript can be published as an article in Nature Communications.

/ Jan Marcus Dahlström

Dear Professor Dahlström,

Thank you once more for your helpful and transparent comments. Your suggestions triggered several considerable modifications of the manuscript that we believe led to a significant strengthening of the manuscript.

Reviewer #2 (Remarks to the Author):

Konrad Hütten et al performed a XUV pump NIR probe study to investigate the electronic dynamics in krypton atoms following XUV excitation. Photo-ion and attosecond transient absorption spectroscopies are combined in a single experiment aiming to explore complex excitation and ionization processes simultaneously. Such combined technique has demonstrated its advantage of extracting multitude of information including the instrument response function, lifetime of autoionizing states as well as the intensity dependence of the multi-step Auger decay processes. After the XUV excitation, the highly excited states $3d-1np$ experiences a two step Auger decay process until the NIR pulse arrives to terminate the relaxation steps. The attosecond transient absorption measurement reveals the lifetime of the first Auger decay process, and the second step Auger decay dynamics is tracked by the photoion measurement. This is an appealing work that holds the promise to access complete dynamic information during light-matter interaction, which is normally inaccessible using single spectroscopy. I recommend its publication in Nature Communication with the following comments addressed.

- 1) Although combining different spectroscopies can potentially lead to richer information, compatibility between different spectroscopies is a prerequisite. For example, the photo-ion/photoelectron spectroscopy prefers dilute target to avoid the space charge effect, a successful combined technique is the Reaction Microscope, which combines the photoelectron and photoion spectroscopy in coincidence. The attosecond transient absorption spectroscopy typically requires a target density that is several orders of magnitude higher than that for the photoelectron/photoion spectroscopy in order to create observable signatures in the transmitted spectrum. This incompatibility indicates that the reported technique is limited to processes that do not have severe ionization, therefore extending the technique to more complex molecular systems with lower ionization threshold, is probably challenging. It will be valuable to include some discussions on this issue.

The Reviewer stumbled over the same weakness of our initially submitted manuscript as pointed out by Reviewer 1, second comment. Indeed, both the dynamic range as well as the desired target density for the two experimental methods are not identical. As the reviewer mentions, the ion mass spectrometry calls for a limited number of ions in the interaction volume to avoid space charge, while the fidelity of the TAS increases with target density. However, both techniques are not only not limited to a certain degree of ionization but in contrast benefit from a large degree of ionization. Even if under certain experimental conditions full ionization would be achievable, for successful ion mass spectrometry it would only be a matter of choosing a proper sample density to limit the number of ions in the detection volume below the threshold for detrimental space charge effects.

As the reviewer suggests we added a brief discussion on this topic to the main text that is overlapping with our response to Reviewer 1, second comment.

We added to lines 53 and following of the original manuscript version:

However, it generally lacks the dynamic range to simultaneously detect the characteristic absorption signals of different co-existing ionic charge states due to insufficient spectral

bandwidth or due to the absorption cross sections and abundances of the different ionization states that typically vary by several orders of magnitude. TAS experiments ...

And to lines 64 and following:

Merging these detection methods provides complementary insight into the excitation/decay mechanism as the transient absorption maps the initiating resonant excitation and the ion detection sensitively records the subsequent branching into intermediate and final states. The combination of transient absorption with ion spectroscopy compensates the limits in the simultaneous detection of several ionic states that absorption spectroscopy typically brings along. The prerequisite is that the pump-probe experiment can be performed twice under identical experimental conditions except for differing target densities optimized for the two detection methods. Adjusting the laser electric field amplitude allows selectively addressing different intermediate states of the autoionization cascades that follow the XUV excitation resulting in different apparent lifetimes. With that, the experimentalist obtains control of the temporal evolution and the absolute yield of the ionization dynamics by accessing different level groups in the cascade with different NIR intensities.

We also expanded the respective SI section in the experimental details:

The background pressure in the experimental chamber without krypton gas load is 7×10^{-8} mbar. For the ion measurement, the krypton gas pressure is homogeneously 3.5×10^{-3} mbar in the experimental chamber while a gas nozzle is introduced to the interaction zone for the TAS measurement resulting in a gas pressure of 5×10^{-3} mbar in the chamber.

- 2) The time constants and the retrieved branching ratio of the Auger decay channels in Fig4b and Fig4c may not be reliable. The humps appearing around 50fs are detrimental for accurate fittings. When humps are getting amplified at higher NIR intensity, it is rather difficult to retrieve a meaningful ratio of A/B. The authors should clarify this since this is the major discovery of the current work.

To address this we added a new section in the SI (section 3) including new experimental data explaining the origin of the signal increase at around 70 fs and proving that its presence is not detrimental for the correct extraction of the time constants and the branching ratios A/B.

NIR pre-pulse causing a transient increase of the Kr^{3+} ion yield at about 70 fs

A common feature of NIR few-cycle laser systems is the appearance of pre-/post-pulses several tens of fs before/after the few-cycle laser pulse due to inevitable oscillations in the group delay dispersion characteristics of the chirped dielectric mirrors that are used to compress the spectrally broadened output of the laser chain. Only in the case of the highest intensity, the accompanying pulse is intense enough to influence the Kr^{3+} ion yield significantly (see figure 4c of the main manuscript). Its presence is revealed by a xenon autocorrelation measurement (see figure S4a).

To account for the transient increase in the Kr^{3+} ion yield at high NIR intensities, the corresponding three elevated data points (marked in red) have been excluded from the fit, resulting in an unaffected extraction of the time constants and branching ratios.

Figure S4: Autocorrelation trace of Xe⁺ (a) and the Kr³⁺ ion yield change presented in the main manuscript for high NIR intensities (b), both vs. time delay. The autocorrelation trace is recorded with interfering two NIR pulse replica of similar intensity in xenon, revealing a pre-pulse 70 fs before the central pulse envelope. This pre-pulse causes a transient increase of the Kr³⁺ ion yield at around 70 fs (b) at the NIR intensity of $(2.9 \pm 0.5) \times 10^{14} \text{ W/cm}^2$. The affected data points (marked red) are not included into the fit.

3) The XUV spectrometer energy resolution is energy dependent; it should be specified at which energy the resolution is 100 meV.

We corrected for that missing piece of information. The text now states a resolution of 100 meV at 90 eV.

4) In line 110 of page 7: the final state for the cascade should be "4s-1 4p-1/4p-34d"

Thank you for finding this mistake. We eliminated it.

Reviewer #3 (Remarks to the Author):

In this manuscript the authors study Auger cascade in Krypton by means of time-resolved techniques, i.e. the Transient Absorption Spectroscopy (TAS) and the photo-ion mass spectroscopy. They found that the TAS gives results in reasonable accordance with the autoionizing process envisioned, however the ion signals indicate the presence of an Auger cascade which lifetime strongly depends on the NIR laser intensity. The combination of these two techniques is rather a "tour de force" and the results are interesting, but I do not think this manuscript deserves publication in Nature Communications.

The introductory comment of reviewer 3 proves that the old version of our manuscript caused a major misunderstanding about the purpose and the conclusion of our experiment: We did not merely try to reproduce transient absorption results by recording ion signals but combined the two complementary techniques. This novel conceptual advance allows to draw for the first time a complete picture of Auger decay processes and enables the control of the ionization dynamics in rare gas atoms. We are convinced that our manuscript deserves publication in Nature Communications because of our new insights about the NIR intensity dependent control of ionization dynamics that have been enabled by the unprecedented alliance of time-resolved absorption and ion spectroscopy and that are supported by theoretical calculations.

In order to eliminate the weakness of our manuscript we made the following changes detailed after each individual comment:

1/ I consider that this manuscript is a follow-up of a paper [13] already published by the same authors. The only differences concern the use of the combined techniques but it is not clear at all, what are the new physical insights gained here.

As we demonstrate, the combination of the two techniques is a huge asset for understanding Auger decay processes. With our presented technique, the ionization dynamics in rare gas atoms can deliberately be influenced. Additionally, our theory work succeeds for the first time to link our experimentally observed decay rates to a weighted average of all decay channels involved rather than to just one decay level as previously reported in [13] (in the revised manuscript cited as [15]). Furthermore, your comment helped us to pinpoint a major misleading conclusion triggered by the way our initial manuscript stated the connection: "Auger cascade which lifetime strongly depends on the NIR laser intensity". The individual lifetimes are not dependent of the NIR intensity. Instead, the variation of the NIR intensity enables the access to different cascade paths that show different behavior regarding the ionization dynamics (i.e., different "apparent" lifetimes). To emphasize these issues in our manuscript, we improved the following paragraphs:

In the introductory paragraph:

In tandem with theory, our study reveals the role of intermediate electronic states in the formation of multiply charged ions. Amplitude tuning of a dressing laser field addresses different groups of decay channels and allows exerting temporal and quantitative control over the ionization dynamics in rare gas atoms.

Lines 64 and following (line numbers always refer to the original manuscript version):

Merging these detection methods provides complementary insight into the excitation/decay mechanism as the transient absorption maps the initiating resonant excitation and the ion detection sensitively records the subsequent branching into intermediate and final states. The combination of transient absorption with ion spectroscopy compensates the limits in the simultaneous detection of several ionic states that absorption spectroscopy typically brings along. The prerequisite is that the pump-probe experiment can be performed twice under identical experimental conditions except for differing target densities optimized for the two detection methods. Adjusting the laser electric field amplitude allows selectively addressing different intermediate states of the autoionization cascades that follow the XUV excitation resulting in different apparent lifetimes. With that, the experimentalist obtains control of the temporal evolution and the absolute yield of the ionization dynamics by accessing different level groups in the cascade with different NIR intensities.

Lines 107 and following:

This value differs from the previously reported value of (20 ± 4) fs¹⁵: Our presented study is to our knowledge the first one that systematically investigates the ionization dynamics in krypton for different NIR intensities. As the lowest NIR intensity is twice as high as previously¹⁵, different autoionization paths in the de-excitation of the Kr⁺ ions become visible and may lead to a different apparent/effective decay time. In practice, we here observe an effective lifetime for the $3d^{-1}np \rightarrow 4s^{-2}np \rightarrow 4s^{-1}4p^{-1}/4p^{-3}4d$ cascades that arise from a large number of individual but not resolved fine-structure transitions. This is confirmed by our multiconfiguration Dirac-Fock calculations using the GRASP²⁷ and RATIP²⁸ codes and by taking the average of all the individual lifetimes, weighted by their relative decay probabilities. This theoretical approach is for the first time applied to reveal effective lifetimes of different potential group cascades, see section 5 in the SI for details.

Additionally, we now emphasize our interaction with different level groups by explicitly stating “effective lifetime” and “apparent lifetime” throughout the manuscript where applicable in order to avoid the misunderstanding of changing the lifetimes themselves – we do not modify lifetimes but make different level groups become visible resulting in different apparent lifetimes.

As a side note, we would like to point out that only one of our authors was involved in the previous work [13], in the revised manuscript version cited as [15].

2/ There is no real answer proposed on the influence of the NIR intensity on the lifetime in the Auger cascade. Different options are presented but I believe the authors should give more detailed reasons.

The NIR intensity does not influence the lifetimes in the Auger cascade. We observe different lifetimes of the Auger cascades because different groups of levels of the Kr⁺ ion become visible with different NIR intensities. While this behavior has already been discussed in the original version of our manuscript, the comment of referee 3 prompted an improvement of the concerning paragraphs. With the revised paragraph starting in line 107, we now emphasize the concept of effective lifetimes of different level groups better than before and refer explicitly to the section in the SI that is exclusively dedicated to the extraction of the effective lifetimes. Despite the overlap with the answer to comment number 1, we repeat here our changes for the sake of the referee’s convenience:

Lines 64 and following:

Merging these detection methods provides complementary insight into the excitation/decay mechanism as the transient absorption maps the initiating resonant excitation and the ion detection sensitively records the subsequent branching into intermediate and final states. The combination of transient absorption with ion spectroscopy compensates the limits in the simultaneous detection of several ionic states that absorption spectroscopy typically brings along. The prerequisite is that the pump-probe experiment can be performed twice under identical experimental conditions except for differing target densities optimized for the two detection methods. Adjusting the laser electric field amplitude allows selectively addressing different intermediate states of the autoionization cascades that follow the XUV excitation resulting in different apparent lifetimes. With that, the experimentalist obtains control of the temporal evolution and the absolute yield of the ionization dynamics by accessing different level groups in the cascade with different NIR intensities.

Lines 107 and following:

This value differs from the previously reported value of (20 ± 4) fs¹⁵: Our presented study is to our knowledge the first one that systematically investigates the ionization dynamics in krypton for different NIR intensities. As the lowest NIR intensity is twice as high as previously¹⁵, different autoionization paths in the de-excitation of the Kr⁺ ions become visible and may lead to a different apparent/effective decay time. In practice, we here observe an effective lifetime for the $3d^{-1}np \rightarrow 4s^{-2}np \rightarrow 4s^{-1}4p^{-1}/4p^{-3}4d$ cascades that arise from a large number of individual but not resolved fine-structure transitions. This is confirmed by our multiconfiguration Dirac-Fock calculations using the GRASP²⁷ and RATIP²⁸ codes and by taking the average of all the individual lifetimes, weighted by their relative decay probabilities. This theoretical approach is for the first time applied to reveal effective lifetimes of different potential group cascades, see section 5 in the SI for details.

3/ Most figures are very difficult to read and should be redesigned to give useful informations to the reader. For example, the choice of yellow in Fig. 1a is « awful » and Figs. 2 cannot be fully exploited (the traces of 82 and 83 Kr³⁺ are invisible). I suggest to remove Figs. 2 from the manuscript and put them in the SI. This would free more space for the authors to give more details.

We improved figures 1 and 2 by the following:

In Figure 1, we changed all parts with yellow and orange shadings into green coloring. In order to avoid an overloaded figure, we moved the autocorrelation trace (panel 1b of the former figure) to the SI section, now being figure S1.

In Figure 2, we changed the scale of the rainbow coloring such that the isotopes 82 and 83 of Kr³⁺ become more visible than before. After a careful reconsideration about the best position of this figure, we decided to keep the figure in the main manuscript because it gives a helpful first overview about our performed measurements. The question of figure design can be a subjective issue. Reviewers 1 and 2 did not complain about the figure design. However, we want to emphasize that the comments of reviewer 3 helped us to improve figures 1, 2 and 4 substantially.

4/ I am very puzzled by the results shown in Fig. 4. It might be an optical illusion but the decrease of Fig.4b seems slower or with the same slope than in Fig.4c, while the ratio A/B are totally different. Could the authors comment on that? Furthermore, I do not understand the notion of post pulse put forward to explain the increase around 80fs. More informations to the reader would be welcome on its origin and why it is not present at lower intensities and what would be the options to remove it.

In figure 4, the incorrect impression of a slower decrease in 4b when compared to 4c is explained by the different scalings of the y axis in all three panels. In order to emphasize this, we included two dashed horizontal lines respectively highlighting the maxima of the fitting functions and the asymptotes that the ion yield relaxes to for long delay times. We also included the difference values of these two important points of the fitting functions on the right side of the figure (violet numbers). Those horizontal lines help to avoid the optical illusion. Additionally, we now explicitly point out the different ion yield scalings for the three NIR intensities in the figure caption:

New figure 4:

Addition to the figure caption: ... For the highest intensity, an accompanying pulse of the ultra-short NIR laser pulse increases the ion yield at around 70-120 fs time delay (see section 3 of the SI for more details). Please note that the scaling, the initial and final values of the ion yields differ significantly for the three NIR intensities. The violet numbers on the right side of each panel indicate the difference between the maximum of the fitting function and the value the fitting function relaxes to for large time delays (see section 4 of the SI for a detailed explanation for the different ion yield values).

There is an own section in the SI (section 4) dealing with the NIR intensity dependent A/B ratio which we modified slightly for better clarity.

Concerning the signal increase at 70-120 fs in figure 4c, we added an own section to the SI (section 3) explaining its origin and including additional experimental data:

NIR pre-pulse causing a transient increase of the Kr^{3+} ion yield at about 70 fs

A common feature of NIR few-cycle laser systems is the appearance of pre-/post-pulses several tens of fs before/after the few-cycle laser pulse due to inevitable oscillations in the group delay dispersion characteristics of the chirped dielectric mirrors that are used to compress the spectrally broadened output of the laser chain. Only in the case of the highest intensity, the accompanying pulse is intense enough to influence the Kr^{3+} ion yield significantly (see figure 4c of the main manuscript). Its presence is revealed by a xenon autocorrelation measurement (see figure S4a).

To account for the transient increase in the Kr^{3+} ion yield at high NIR intensities, the corresponding three elevated data points (marked in red) have been excluded from the fit, resulting in an unaffected extraction of the time constants and branching ratios.

Figure S4: Autocorrelation trace of Xe^+ (a) and the Kr^{3+} ion yield change presented in the main manuscript for high NIR intensities (b), both vs. time delay. The autocorrelation trace is recorded with interfering two NIR pulse replica of similar intensity in xenon, revealing a pre-pulse 70 fs before the central pulse envelope. This pre-pulse causes a transient increase of the Kr^{3+} ion yield at around 70 fs (b) at the NIR intensity of $(2.9 \pm 0.5) \times 10^{14} \text{ W/cm}^2$. The affected data points (marked red) are not included into the fit.

5/ The authors claim they have a coherent control on the decay channels but I would not call it « coherent » as it does not depend on the delay between the XUV and the NIR nor on the NIR phase. It is only a « intensity » effect, from my point of view.

Indeed, “coherent control” usually refers to the manipulation of interference phenomena by phase shaping laser pulses rather than influencing these phenomena with a coherent light source. In order to avoid this misunderstanding we deleted the term “coherent” in the figure caption 4.

REVIEWERS' COMMENTS:

Reviewer #1 (Remarks to the Author):

The authors have responded well to all my major points. The new version is more clear and much improved concerning major findings and fusion of experimental techniques. I think that the manuscript can be published in Nature Communications now without further revisions required.

Reviewer #2 (Remarks to the Author):

The authors have addressed all my previous comments and made proper corrections, I thus recommend its publication in Nature Communication.

Reviewer #3 (Remarks to the Author):

In the new version of the manuscript, the intent of the authors are much clearer and as they answered to all questions raised by the referees, I am now convinced that it deserved publication in Nature Communications.